# Human-Prosthetic Interaction (HumanIT): A study protocol for a clinical trial evaluating brain neuroplasticity and functional performance after lower limb loss

Elke Lathouwers[1,2], Bruno Tassignon[1], Alexandre Maricot[1], Ahmed Radwan[3], Maarten Naeyaert[4], Hubert Raeymaekers[4], Peter Van Schuerbeek[4], Stefan Sunaert[3,5], Johan De Mey[4], Kevin De Pauw[1,2,6] *

1 Human Physiology and Sports Physiotherapy research group, Vrije Universiteit Brussel, Brussels, Belgium, 2 BruBotics, Vrije Universiteit Brussel, Brussels, Belgium, 3 KU Leuven, Department of Imaging and pathology, Translational MRI, Leuven, Belgium, 4 Department of Radiology and Magnetic Resonance, UZ Brussel, Jette, Belgium, 5 UZ Leuven, Department of Radiology, Leuven, Belgium, 6 Strategic Research Program 'Exercise and the Brain in Health & Disease: The Added Value of Human-Centered Robotics', Vrije Universiteit Brussel, Brussels, Belgium

* Kevin.De.Pauw@vub.be

**Data Availability Statement:** No datasets were generated or analysed during the current study. All

## Abstract

### Background

Lower limb amputation contributes to structural and functional brain alterations, adversely affecting gait, balance, and overall quality of life. Therefore, selecting an appropriate prosthetic ankle is critical in enhancing the well-being of these individuals. Despite the availability of various prostheses, their impact on brain neuroplasticity remains poorly understood.

### Objectives

The primary objective is to examine differences in the degree of brain neuroplasticity using magnetic resonance imaging (MRI) between individuals wearing a new passive ankle prosthesis with an articulated ankle joint and a standard passive prosthesis, and to examine changes in brain neuroplasticity within these two prosthetic groups. The second objective is to investigate the influence of prosthetic type on walking performance and quality of life. The final objective is to determine whether the type of prosthesis induces differences in the walking movement pattern.

### Methods

Participants with a unilateral transtibial amputation will follow a 24-week protocol. Prior to rehabilitation, baseline MRI scans will be performed, followed by allocation to the intervention arms and commencement of rehabilitation. After 12 weeks, baseline functional performance tests and a quality of life questionnaire will be administered. At the end of the 24-week period, participants will undergo the same MRI scans, functional performance tests

relevant data from this study will be made available upon study completion.

**Funding:** The funders did not and will not have a role in study design, data collection and analysis, decision to publish, or preparation of the manuscript." And we would like to add that "The authors received no specific funding for this work.

**Competing interests:** The authors have declared that no competing interests exist.

and questionnaire to evaluate any changes. A control group of able-bodied individuals will be included for comparative analysis.

## Conclusion

This study aims to unravel the differences in brain neuroplasticity and prosthesis type in patients with a unilateral transtibial amputation and provide insights into the therapeutic benefits of prosthetic devices. The findings could validate the therapeutic benefits of more advanced lower limb prostheses, potentially leading to a societal impact ultimately improving the quality of life for individuals with lower limb amputation.

## Trial registration

NCT05818410 (Clinicaltrials.gov).

## Introduction

A lower limb amputation is a life-altering event that significantly affects an individual's overall well-being or quality of life due to the loss of mobility, physical strains and psychosocial impacts [1, 2]. The absence of a limb restricts the individual's ability to perform daily tasks, leading to physical discomfort, secondary injuries and a higher risk of falling [3–7]. A recent review indicates that 54% of the people with a lower limb amputation wearing a standard passive prosthesis reported falling in the last year, with 26% reporting falling multiple times [7]. On top, it is estimated that 60% of these falls require medical attention [8, 9]. Falls can result in additional injuries, further impairing mobility and independence and increasing fall-related healthcare costs [7, 9, 10]. Addressing these challenges is crucial to improving overall well-being and enhancing the individual's quality of life.

The brain is often overlooked when discussing the effects of amputation. Even though, after an amputation, significant structural and functional neuroplastic changes occur, contributing to decreased static and dynamic balance, increased gait variability and an increased **risk of falling** [7, 11, 12]. Gait patterns and the onset of falls are orchestrated by the intricate interplay between biomechanical factors and the human brain [13]. This interaction entails that the brain plays a vital role in the organisation and performance of human gait [13]. Magnetic resonance imaging (MRI) revealed that lower limb amputation contributes to the thinning of the premotor cortex and visual-motor area, the reduction in white matter integrity within the premotor region contralateral to the amputation site, along with changes in the connectivity between bilateral premotor cortices [12]. These changes disrupt the processes involved in movement planning and the coordination of eye movements relative to the limbs, resulting in reduced perception-action coupling [12]. In addition, amputation influences changes in limb representation in the primary motor and somatosensory cortex. It induces decreased connectivity in the primary motor cortex, primary somatosensory cortex, supplementary motor area, basal ganglia, thalamus, and cerebellum [12]. These changes in connectivity translate towards reduced motor control and balance, further complicating the performance of daily activities of people with lower limb amputation [12].

Given the challenges posed by these neurological changes, enhancing the quality of life for people with amputation necessitates the use of suitable prosthetic ankle devices. Today, passive devices are among the prosthetic feet most widely used [14]. The majority of passive mechanical prostheses rely on a fixed spring, are not articulated and offer basic functionality [14]. To

better mimic able-bodied gait, articulated ankle joints are integrated into such passive devices, enhancing biomechanical factors and increasing the quality of life (e.g. increased mobility, comfort and gait patterns) [14–16]. Yet, there is insufficient evidence for their beneficial influence on brain neuroplasticity [17].

Understanding the influence of passive and articulated passive prostheses on brain neuroplasticity will provide vital information on the underlying mechanisms in the context of fitting an individual with amputation with the most beneficial prosthetic device to improve functionality, reduce fall risk and ultimately improve quality of life. A recent systematic review compared the therapeutic benefits of different ankle prostheses (i.e., passive, quasi-passive and active) during daily activities [17]. This review demonstrated that although many short-term benefits of advanced prosthetic devices have been shown through user evaluations, there is a lack in the literature concerning mid-to-long-term holistic assessments employing a psychophysiological approach [17]. Furthermore, among the included manuscripts, only De Pauw et al. investigated the difference in brain dynamics between two types of prosthetic ankles by means of EEG [18]. Nevertheless, research on the impact of the prosthesis type on neuroplasticity is notably absent from scientific literature. Therefore, our study aims to research brain neuroplasticity induced by different types of prosthetic ankle devices.

## Objectives

The primary objective is to examine the differences in the degree of brain neuroplasticity using magnetic resonance imaging (MRI) between individuals wearing a new passive ankle prosthesis with an articulated ankle joint and a standard passive prosthesis, and to examine changes in brain neuroplasticity within these two prosthetic groups. We hypothesise that the neuroplastic changes influenced by the amputation will be less pronounced in people using an articulated prosthesis than those using a standard passive device. The secondary objective is to investigate walking performance and quality of life related to the type of prosthesis. We hypothesise that walking performance and quality of life will increase when using the prosthesis with an articulated ankle joint. The final objective is to determine whether the type of prosthesis induces differences in the walking movement pattern in individuals with a unilateral lower limb amputation. We hypothesise that movement patterns will more closely align with able-bodied gait patterns in individuals wearing the prosthesis with an articulated ankle joint compared to those wearing the standard passive device.

## Materials and methods

The study protocol has been reported in compliance with the SPIRIT 2013 guidelines [19]. The World Health Organization trial registration data is presented in S1 Table. The completed SPIRIT 2013 checklist and the original protocol are also provided as supporting information.

### Study design & sample size

The study has been designed as a controlled clinical trial with three parallel study arms. Given the nature of the research involving distinct prostheses, neither participant nor investigator blinding is possible. Fig 1 shows a diagram with the different phases of the study. Participants with a unilateral transtibial amputation will be allocated to one of the two prosthetic study arms (i.e. standard care prosthesis vs new articulated prosthesis) with alternate (1:1) allocation. The primary endpoints of the clinical trial are the differences in brain neuroplasticity and the distance covered during the 6-minutes' walk test between the prosthetic arms. With this study being the first to investigate brain neuroplasticity in relation to the type of prosthesis, data to conduct sample size estimation is lacking. Therefore, the sample size for this study was

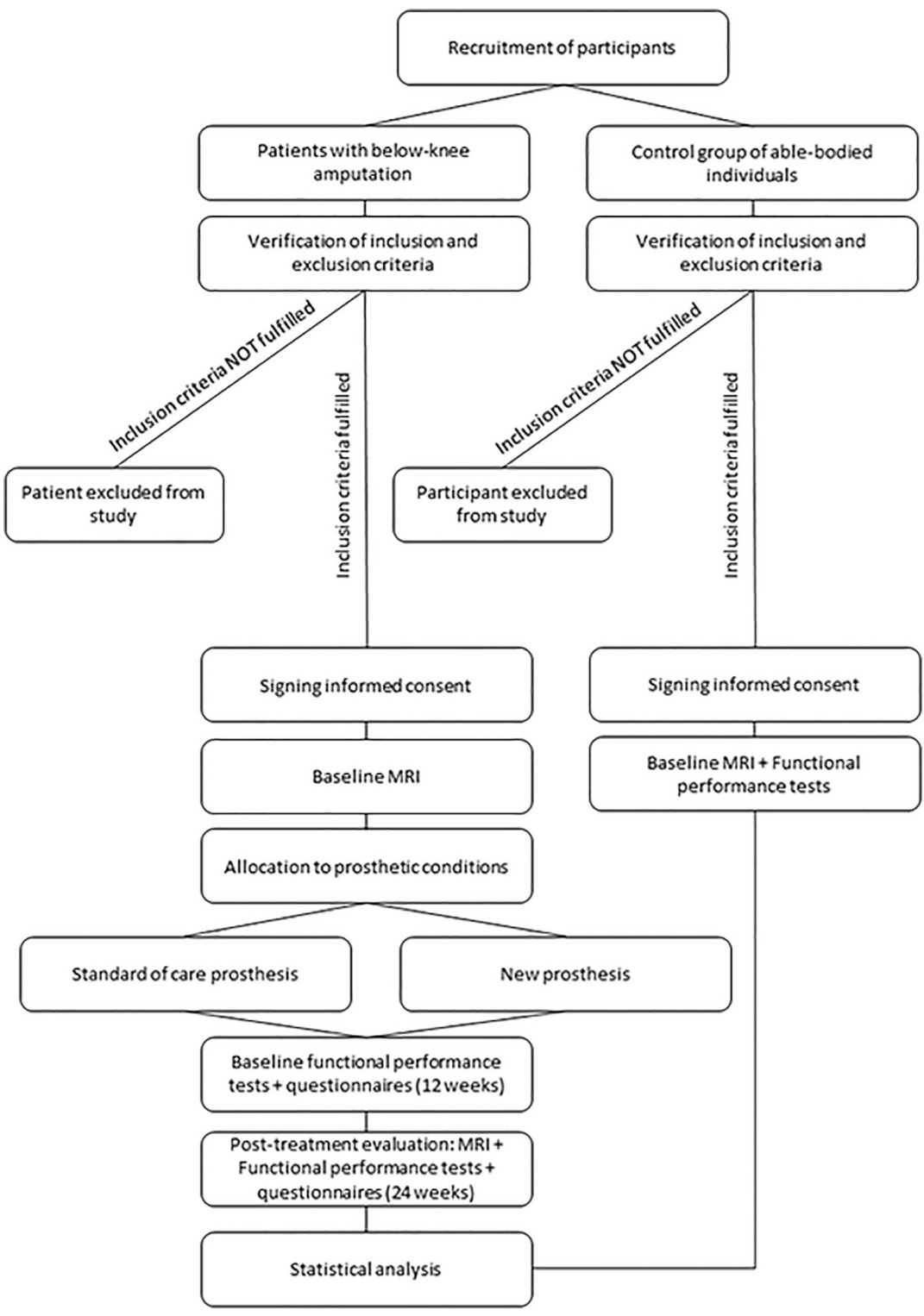

**Fig 1. Study flowchart.**

calculated based on the data available for the distance covered during the 6-minutes' walk test [20]. We performed a sample size calculation for an analysis of variance using G*Power (version 3.1.9.4) envisaging 10% drop-out. Values used for this calculation were medium expected effect size f = 0.318, power = 0.80, alfa = 0.05, the number of groups = 3 and the number of measurements set at 2 with a correlation between measurements of 0.75. A total of 40 participants with a lower limb amputation is required and will be allocated alternatingly (i.e., pseudo-randomisation) to one of the two prosthetic groups. A control-group of 20 able-bodied individuals will be included to a third study arm to enable comparison.

## Inclusion and exclusion criteria

Adults aged 25–65 with a unilateral transtibial amputation will be recruited through four rehabilitation centres, and orthopaedic departments of hospitals in Belgium. Recruitment will take place from September 2023 onwards until the required sample size is attained. Adults with a Medicare Functional Classification level < K3, metal implants, bilateral amputation, additional upper-limb amputation or diabetes and participants with neurological disorders or excessive stump pains and wounds will be excluded. All participants will be asked to provide their written consent after being written and verbally informed regarding the study protocol. The study will be executed in compliance with the Declaration of Helsinki [21] and is approved by the medical ethics commission of VUB and UZ Brussel (BUN 1432023000077). The study is registered via Clinical.trials.gov (NCT05818410).

## Protocol

We will compare a new passive articulated prosthetic ankle-foot (Lunaris®) to a conventional non-articulated prosthetic foot (SACH®) on brain neuroplasticity, functional physical performance, quality of life and movement patterns by means of a clinical trial.

The clinical trial will comprise four test days for participants with a lower limb amputation and two days for the control group of able-bodied individuals. The able-bodied individuals will undergo an MRI-scan at the University hospital Brussels followed by functional performance tests conducted.

Participants with a lower limb amputation will start the clinical trial upon the start of their rehabilitation. The rehabilitation will be provided by the patients their physiotherapist of choice and is not part of this trial. At week 0, when initiating the rehabilitation, participants will undergo a baseline MRI-scan at the University hospital Brussels. Then, they will be allocated alternatingly to the intervention arms (new articulated ankle-foot prosthesis or standard foot prosthesis) and will conduct their rehabilitation to learn walking with a prosthesis. At the end of the rehabilitation, at 12 weeks, participants will perform baseline functional performance tests, and fill out the prosthetic evaluation questionnaire measuring quality of life [22]. Between weeks 12 and 24 of the clinical trial, participants will perform their daily activities with their allocated prosthesis. During this 12-week period, trying out new prosthetic devices will be allowed within the group of individuals wearing the standard prosthetic foot as this is considered the usual care. At the end of this period (at week 24), post-intervention assessments will take place and participants will undergo the same MRI and functional performance tests and fill out the same quality of life questionnaire to evaluate the changes that occurred. Additionally, participants will be asked to fill out the Quebec User Evaluation of Satisfaction with Assistive Technology to evaluate their prosthetic satisfaction [23, 24]. The 12-week intervention period is chosen based on studies examining the effect of 12 weeks balance training in healthy and older adults on neuroplasticity and on the accommodation time to walking with a new prosthesis [25–27]. Throughout the study, the time during which the individuals walked

| TIMEPOINT** | STUDY PERIOD | | | | |
| --- | --- | --- | --- | --- | --- |
| | Enrolment | Allocation | Post-allocation | | |
| | - Week 0 | Week 0 | Week 0 | Week 12 | Week 24 |
| **ENROLMENT** | | | | | |
| Eligibility screen | X | | | | |
| Informed consent | X | | | | |
| Allocation | | X | | | |
| **TREATMENT** | | | | | |
| New articulated passive ankle-foot prosthesis | X | X | X | X | X |
| Standard foot prosthesis | X | X | X | X | X |
| Able-bodied individuals | X | ■ | X | ■ | ■ |
| **ASSESSMENTS** | | | | | |
| Brain neuroplasticity | | | X | ■ | X |
| Quality of life | | | ■ | X | X |
| Prosthetic satisfaction | | | ■ | ■ | X |
| Functional performance & questionnaires | | | X* | X | X |

**Fig 2. The schedule of enrolment, interventions, and assessments.** *only for able-bodied individuals.

with their prosthetic device over the 24h period will be estimated based on anamnestic questions on the average hours of prosthetic use a day, duration and frequency of rehabilitation period. Fig 2 presents an overview of the schedule of enrolment, interventions, and assessments.

## Auxiliary subject information and outcome parameters

An overview of the different outcome parameters is provided in Table 1.

**Auxiliary subject information.** Auxiliary subject information will be collected during the first day of data collection for the participants with an amputation. The subject information that will be collected comprises age, sex, height, length, weight, handedness, average amount of daily physical activity, date of amputation, side of amputation, stump length, comorbidities,

**Table 1. Outcome parameters.**

| | | |
|---|---|---|
| **Primary outcomes** | | |
| Brain neuroplasticity | MRI: brain connectivity & microstructure | [numeric] |
| Functional performance | | |
| 6-min walking test [43] | Distance covered in 6 minutes | [numeric] |
| **Secondary outcomes** | | |
| Functional performance | | |
| Slope walking test [40] | Time to complete test | [numeric] |
| Stair climbing and descending test [41] | Time to complete test | [numeric] |
| Dual-task L-test [42] | Time to complete test & accuracy of subtractions | [numeric] |
| L-test [42] | Time to complete test | [numeric] |
| 6-min walking test [43] | Time to complete test | [numeric] |
| Biomechanical gait parameters | Lower limb joint angles and velocities & intra-limb continuous relative phases | [numeric] |
| Slope walking test [40] | Lower limb joint angles and velocities & intra-limb continuous relative phases | [numeric] |
| Stair climbing and descending test [41] | Lower limb joint angles and velocities & intra-limb continuous relative phases | [numeric] |
| 6-min walking test [43] | Lower limb joint angles and velocities & intra-limb continuous relative phases | [numeric] |
| Scores | | |
| Prosthetic Evaluation Questionnaire | | [numeric, VAS, scale] |
| Quebec User Evaluation of Satisfaction with Assistive Technology | | [numeric, scale] |
| NASA-Task-Load Index | | [numeric, VAS] |
| L-test [42] | | [numeric, VAS] |
| Dual-task L-test [42] | | [numeric, VAS] |
| Level of comfort | | [numeric, VAS] |
| Slope walking test [40] | | [numeric, VAS] |
| Stair climbing and descending test [41] | | [numeric, VAS] |
| Dual-task L-test [42] | | [numeric, VAS] |
| L-test [42] | | [numeric, VAS] |
| 6-min walking test [43] | | [numeric, VAS] |
| Level of fatigue | | [numeric, VAS] |
| Slope walking test [40] | | [numeric, VAS] |
| Stair climbing and descending test [41] | | [numeric, VAS] |
| Dual-task L-test [42] | | [numeric, VAS] |
| L-test [42] | | [numeric, VAS] |
| 6-min walking test [43] | | [numeric, VAS] |
| Rate of perceived exertion | | [numeric] |
| Slope walking test [40] | | [numeric] |
| Stair climbing and descending test [41] | | [numeric] |
| Dual-task L-test [42] | | [numeric] |
| L-test [42] | | [numeric] |
| 6-min walking test [43] | | [numeric] |

MRI = Magnetic Resonance Imaging, VAS = Visual Analogue Scale

use of medication, presence of phantom pains, history of trauma and the history of falls. During the follow-up measurements, we will again perform the anamnesis and extend this with questions on the rehabilitation process, if other prosthetic devices were tested, and the duration of prosthesis use throughout the day. Auxiliary subject information that will be gathered during the first day of data collection within the control group of able-bodied participants will comprise age, sex, height, length, weight, handedness and average daily physical activity.

**Brain neuroplasticity.**   Neuroplasticity can be defined as a reorganisation of brain structure, function and connectivity within the central nervous system in response to intrinsic and extrinsic stimuli [28]. Based on our hypotheses, we are mainly interested in, but not limited to, the motor-related regions of the brain given the exploratory nature of this study [12]. The outcome parameters include connectivity and white and grey matter microstructure measures (e.g., characteristic path length, clustering coefficient, efficiency, neurite density index, fiber density, and fiber cross-section, fiber density cross-section).

Pre- and post-intervention MRI scans will be acquired supine on a 3-Tesla MRI unit using a 48-channel head coil. A 30-minute protocol will be followed, comprising scout images, 3D T1-weighted spin-echo images, diffusion-weighted imaging, resting state-fMRI, and 3D-QALAS. An anatomical 3D T1-weighted image and a multi-shell diffusion-weighted dataset will be acquired, as well as a resting-state fMRI acquisition and a 3D-QALAS acquisition. The diffusion dataset includes several b0 images, and for both the diffusion and resting state data, additional data using reversed-phase directions will be acquired. The 3D-QALAS acquisition allows for a simultaneous T1, T2 and proton density mapping, additional data using reversed-phase directions will be acquired, making it sensitive to microstructural changes in the brain [29, 30].

After data acquisition, the MRI data will be pre-processed, and the outcome parameters will be computed. FSL will be used to pre-process the diffusion-weighted imaging and resting state-fMRI data [31–33]. Pre-processing consists of denoising, distortion, eddy current and bias field correction, and motion correction with outlier replacement. Afterward, a MRtrix3 performs a fixed-based analysis in the white matter regions, while a NODDI analysis is done in the grey matter areas [34, 35]. To analyse the 3D-QALAS data, SyMRI will be used (SyntheticMR AB, Linköping, Sweden), after which grey matter, white matter and myelin maps are estimated from the relaxometry maps [36, 37]. Once the relaxometry and tissue maps are obtained, a voxel-based analysis will be performed using a modified version of the hMRI toolbox of SPM12 [38, 39], using MATLAB.

**Functional performance & questionnaires.**   A 6-min walking test, dual-task L-test, L-test, slope walking test, and stairs climbing and descending test will indicate functional balance and dual-task performance [40–43]. Rating of perceived exertion [44], level of fatigue and comfort [45] and NASA-Task Load index (NASA-TLX) assessing perceived workload [46] will be assessed after each task.

**Quality of life & prosthetic satisfaction.**   The prosthetic evaluation questionnaire will be used to assess the quality of life [22]. The Quebec User Evaluation of Satisfaction with Assistive Technology will be used to evaluate the prosthetic satisfaction and prosthetic associated services [23, 24].

**Biomechanical gait parameters.**   Inertial measurements units (Awinda, Xsens Technologies B.V., Enschede, The Netherlands) will be capturing 3D-accelerations and angular velocities during the 6-min walking test, the slope walking test and the stairs climbing and descending test to evaluate the participants' gait pattern [47]. Inertial measurement units will be placed on the feet, lower legs, upper legs and pelvis according to the manufacture guidelines. The data of the inertial measurement units will be used to evaluate coordination patterns between hip, knee and ankle joints or segments based on continuous relative phase plots.

Biomechanical data will be analysed and purified in Matlab (The MathWorks Inc., Massachusetts, United States) following the methods described in Lathouwers et al (2023) [48].

## Statistical analyses

Statistical analyses will be performed using R [49], MRtrix [50], and the hMRI toolbox [51]. Statistical methods for brain neuroplasticity and functional performance testing outcomes will include mixed linear or generalised linear modelling depending on the model assumptions being fulfilled. If those analyses cannot be conducted, assumptions will be checked to perform multivariate or univariate analyses. If the assumptions for those types of analyses are also not fulfilled, non-parametric equivalents will be applied. The data will be adjusted for confounding variables like age, sex, handedness, amount of daily prosthetic use and the average amount of daily physical activity. The biomechanical gait parameters will be compared within and between groups using statistical parametric or non-parametric mapping. The statistical significance level will be set at 5% for all analyses.

## Data storage and management

As personal and sensitive data of human participants will be collected, pseudonymisation of the data will occur as soon as possible upon data acquisition. The file, which establishes the link between the pseudonymised identities and the participant's identification, will be password-protected.

Raw data from the anamnesis, functional performance tests and questionnaires will be collected via an electronic case report form using Redcap (projectredcap.org). Data collected in Redcap is automatically stored on the external server of the Ethical Committee of the UZ Brussel. Upon completion of the data collection, the data collected via Redcap will be downloaded from the software application and used for data processing and analyses. Both the raw and cleaned data will be stored on the research's group network attached storage (NAS). Raw biomechanical and MRI data will be stored on the research's group network attached storage (NAS). After data processing, the cleaned data will be stored on the NAS. The NAS is system-encrypted, back-upped, has up-to-date antivirus and is only accessible by authorised personnel via two-factor authentication. Manual files will be stored in locked filing cabinets. All pseudonymized data will be stored for up to 10 years, as recommended by the Vrije Universiteit Brussel.

Data in the present study will be managed confidentially and conform to the General Data Protection Regulation (GDPR) of 27 April 2016. Participating in this study means that the participant agrees that the investigators gather data of the participant that will be used to conduct the investigation and eventually will result in publications in scientific journals and presentations at conferences. By signing the informed consent, participants also provide their agreement that encoded data may be shared with other researchers beyond the scope of the current study. The participant will always have the right to ask the investigators which data he/she has gathered on them and for what this data is used in light of the investigation. The participant retains the right to look into this data and to ask for corrections on the occasion this data contains errors. The participant also retains the right to stop his/her participation in the study at any moment.

In order to control the quality of the present study, the medical file of the participants may be consulted. If so, this will be done by qualified personnel bound to professional secrecy, e.g. representatives of the ethical committee or an external auditing bureau. This consultation can only occur under strict conditions, under the investigator's responsibility and supervision. The

encrypted data can also be passed through the Belgium government, other regulatory authorities, or the ethical committee.

## Ethics and dissemination

The study is approved by the medical ethics commission of VUB and UZ Brussel (BUN 1432023000077), and is registered via Clinical.trials.gov (NCT05818410). Written informed consent will be obtained from all participants.

The authors of the present protocol proposal declare no financial or other conflicts of interest.

Trial results will be communicated to the general public through the publication of disseminated manuscripts published in scientific journals, and through conference presentations/posters. Authorship of these papers and presentations will only be granted to researchers that performed a substantial contribution to the scientific article [52]. Given the nature of this paper (i.e. protocol paper), data has yet to be gathered. After data collection and analysis, the statistical code and dataset will be freely available in a to-be-determined online repository for lasting access possibility.

## Discussion & conclusion

This study will examine differences in brain neuroplasticity through MRI between individuals wearing a new passive ankle prosthesis with an articulated ankle joint and a standard passive ankle prosthesis. The secondary aim is to investigate walking performance and quality of life-related to the type of prosthesis, and the final aim is to determine if the type of prosthesis changes the movement pattern during walking.

This novelty of the study stems from it being the first to unravel possible differences in neuroplasticity in relation to the type of prosthesis in patients with a lower-extremity amputation. It could provide valuable insights into how the brain adapts to various prosthetic types, complementing the existing literature on neuroplasticity after amputation [12]. Depending on the findings of this study, these results may have a societal impact and improve the quality of life for individuals with lower limb amputation.

Furthermore, this study will generate biopsychosocial findings on the therapeutic health benefits of prostheses on individuals with an amputation (i.e. functional performance and quality of life) through both subjective and objective parameters. The study will, therefore, take a first step in filling the literature gap regarding medium- to longer-term prosthetic effects on the individual wearing the prosthesis [15]. The findings from this study may serve as a stepping stone for future research, fostering a deeper understanding of the overall impact of prosthetic interventions. Additionally, they have the potential to inform future advancements in prosthetic design and rehabilitation strategies, aiming to improve the quality of life of patients with an amputation and decrease the associated mortality risk.

## Supporting information

**S1 Table. Trial registration data.**
(DOCX)

**S1 Protocol. Original protocol.**
(PDF)

**S1 Checklist. SPIRIT 2013 checklist: Recommended items to address in a clinical trial protocol and related documents\*.**
(DOC)

## Acknowledgments

We thank the company Axiles Bionics for their willingness to provide the prosthetic ankle-feet (Lunaris®) for conducting this study and to provide the premises for conducting the functional performance testing. We thank Dr. Dirk Claes, Dr. Christophe Lafosse & Carola Vanwalle from Revarte (Antwerp, Belgium), Dr. Sybille Geers from UZ Gent (Gent, Belgium), Dr. Eva Duinslaeger from UZ Leuven Pellenberg (Pellenberg, Belgium), and Dr. Antoine Royer and Dr. Kambiz Minooee from Centre de Traumatologie et de Réadaptation (Brussels, Belgium) for their willingness to recruit patients.

## Author Contributions

**Conceptualization:** Elke Lathouwers, Bruno Tassignon, Kevin De Pauw.

**Funding acquisition:** Kevin De Pauw.

**Methodology:** Elke Lathouwers, Bruno Tassignon, Alexandre Maricot, Ahmed Radwan, Maarten Naeyaert, Hubert Raeymaekers, Peter Van Schuerbeek, Stefan Sunaert, Johan De Mey.

**Resources:** Maarten Naeyaert, Hubert Raeymaekers, Peter Van Schuerbeek, Stefan Sunaert, Johan De Mey, Kevin De Pauw.

**Software:** Ahmed Radwan, Maarten Naeyaert, Hubert Raeymaekers, Peter Van Schuerbeek, Stefan Sunaert, Johan De Mey.

**Supervision:** Kevin De Pauw.

**Writing – original draft:** Elke Lathouwers.

**Writing – review & editing:** Elke Lathouwers, Bruno Tassignon, Alexandre Maricot, Ahmed Radwan, Maarten Naeyaert, Hubert Raeymaekers, Peter Van Schuerbeek, Stefan Sunaert, Johan De Mey, Kevin De Pauw.

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
