## [Decision Letter · Decision Letter 0]

14 Dec 2023

PONE-D-23-27271Human-Prosthetic Interaction: A study protocol for a clinical trial evaluating brain neuroplasticity and functional performance after lower limb lossPLOS ONE

Dear Dr. De Pauw,

Thank you for submitting your manuscript to PLOS ONE. After careful consideration, we feel that it has merit but does not fully meet PLOS ONE’s publication criteria as it currently stands. Therefore, we invite you to submit a revised version of the manuscript that addresses the points raised during the review process.

We look forward to receiving your revised manuscript.

Kind regards,

Heike Vallery

Academic Editor

PLOS ONE

Journal Requirements:

Therapeutic benefts of lower limb prostheses: a systematic review - https://doi.org/10.1186/s12984-023-01128-5

In your revision ensure you cite all your sources (including your own works), and quote or rephrase any duplicated text outside the methods section. Further consideration is dependent on these concerns being addressed.

"The funders did not and will not have a role in study design, data collection and analysis, decision to publish, or preparation of the manuscript."

a) If there are ethical or legal restrictions on sharing a de-identified data set, please explain them in detail (e.g., data contain potentially sensitive information, data are owned by a third-party organization, etc.) and who has imposed them (e.g., an ethics committee). Please also provide contact information for a data access committee, ethics committee, or other institutional body to which data requests may be sent. Please note that authors, including Corresponding Authors, are not permitted to be the sole point of contact for data requests.

b) If there are no restrictions, please provide the minimal anonymized data set necessary to replicate your study findings as either Supporting Information files or to a stable, public repository and provide us with the relevant URLs, DOIs, or accession numbers. For a list of acceptable repositories, please see http://journals.plos.org/plosone/s/data-availability#loc-recommended-repositories.

5. Please amend either the title on the online submission form (via Edit Submission) or the title in the manuscript so that they are identical.

7. We note that the original protocol file you uploaded contains a confidentiality notice indicating that the protocol may not be shared publicly or be published. Please note, however, that the PLOS Editorial Policy requires that the original protocol be published alongside your manuscript in the event of acceptance. Please note that should your paper be accepted, all content including the protocol will be published under the Creative Commons Attribution (CC BY) 4.0 license, which means that it will be freely available online, and any third party is permitted to access, download, copy, distribute, and use these materials in any way, even commercially, with proper attribution.

Therefore, we ask that you please seek permission from the study sponsor or body imposing the restriction on sharing this document to publish this protocol under CC BY 4.0 if your work is accepted. We kindly ask that you upload a formal statement signed by an institutional representative clarifying whether you will be able to comply with this policy. Additionally, please upload a clean copy of the protocol with the confidentiality notice (and any copyrighted institutional logos or signatures) removed.

**Additional Editor Comments:**

Please carefully address all reviewers' comments by modifications in the manuscript. Importantly, the data policy appears currently not in line with PLOS ONE requirements. For example, deidentified data to reproduce paper results should be available publicly and stored in a public repository with a persistent DOI. If data is persistently available, this apparently contradicts the stated retention period of 10 years. However, this discrepancy may be due to the formulations in the manuscript and vagueness in specifying types of data. The manuscript needs to more clearly specify all different types of data originally collected or generated from processing, which consent will be asked from participants for each specific type of data, and which data will be made available where. Please carefully consider the PLOS One instructions on data: https://journals.plos.org/plosone/s/data-availability.

Reviewers' comments:

Reviewer's Responses to Questions

**Comments to the Author**

1. Does the manuscript provide a valid rationale for the proposed study, with clearly identified and justified research questions?

Reviewer #1: Yes

Reviewer #2: Yes

2. Is the protocol technically sound and planned in a manner that will lead to a meaningful outcome and allow testing the stated hypotheses?

Reviewer #1: Partly

Reviewer #2: Partly

3. Is the methodology feasible and described in sufficient detail to allow the work to be replicable?

Reviewer #1: No

Reviewer #2: No

4. Have the authors described where all data underlying the findings will be made available when the study is complete?

Reviewer #1: No

Reviewer #2: Yes

5. Is the manuscript presented in an intelligible fashion and written in standard English?

Reviewer #1: Yes

Reviewer #2: Yes

6. Review Comments to the Author

You may also provide optional suggestions and comments to authors that they might find helpful in planning their study.

Reviewer #1: PONE-D-23-27271: statistical review

SUMMARY. This is a study protocol for a clinical trial that will evaluate brain neuroplasticity and functional performance after lower limb loss. The research questions are clearly stated and look sound to me, the sample size computations rely on reasonable assumptions, limitations of the study are listed. My overall impression is therefore positive. However, I list below some points that should be clarified.

SPECIFIC POINTS

1) Allocation to treatment arms. It is not clear whether allocation to treatments will be random or not. Please clarify.

2) Data permissions. It is not clear whether participants will be asked to agree on making deidentified data available to the public (see also point no. 3).

3) Data availability. It is good, and consistent with the policy of the journal, that the authors wish to make deidentified data publicly available. That the data managers would reserve the right to refuse data access is instead a bit too generic. The authors should clarify the criteria that will be used to refuse access to the data. The public repository where the data will be stored should also be declared.

4) Statistical analysis. The sentence “methods … will include multivariate and univariate analyses and t-tests” is a bit too generic. First, the longitudinal structure of the data should be accounted for (data will be in the form of repeated measurements). Second, differences will be affected by confounding factors (age, gender, BMI, …) and will need to be adjusted (see also point no 5). As a result, mixed-effects regression models should be considered to analyze the data.

5) While the list of primary and secondary outcomes is well documented, the protocol lacks information about auxiliary subject information (age, gender, BMI, …) that will be collected.

Reviewer #2: This study protocol outlines the experimental design for a clinical trial aimed at evaluating the neuroplastic, functional, quality of life, and biomechanical effects associated with the use of lower limb transtibial prostheses featuring a passive articulated ankle. The results will be compared to control cohorts, one using a standard SACH-style ankle and another comprising an able-bodied group. Overall, the manuscript is well-written, and the experimental design is sound. However, I have a few concerns that should be addressed prior to consideration for publication.

1) While the authors provide evidence that the neuroplastic changes related to prosthetic ankle choice remain poorly understood, it was not clear to me how novel the functional, quality of life, and biomechanical assessments are. I suggest that the authors provide further discussions around the existing literature and describe how the data collected in their two secondary aims will contribute to the knowledge in the field. I recommend considering a more focused approach, such as evaluating correlations across numerous outcome parameters. Might I suggest that the author consider restricting their aims to a single a, perhaps evaluating correlations across their numerous outcome parameters. For example, the objective of the protocol may be stronger if it were about examining neuroplasticity and how these changes may relate to functional, QoL, and biomechanical outcomes. It certainly would be interesting to better understand how biomechanics and neuroplasticity may be related, etc.

2) The MRI analysis is central to the first objective of this study. However, when describing this outcome measure (line 186), there is very little information about how the collected MRI data will be analyzed. I suggest that the authors further develop and describe a plan for the handling and analysis of MRI data. For instance, they could elaborate on the specific measures, brain regions, networks, temporal changes, etc., that will be interrogated, and clarify how they plan to quantify the degree of plasticity demonstrated.

3) Lines 103-106 briefly describe a prior systematic review. It would strengthen the manuscript to summarize what the literature highlighted in this section found and how it relates to this study protocol.

7. PLOS authors have the option to publish the peer review history of their article (what does this mean?). If published, this will include your full peer review and any attached files.

Reviewer #1: No

Reviewer #2: **Yes: **Jonathon Schofield

---

## [Author Response · Author response to Decision Letter 0]

25 Jan 2024

Response to the Editor and Reviewers

Dear editor and reviewers, 

We are pleased to send you the revised version of our manuscript (Submission ID PONE-D-23-27271) entitled “Human-Prosthetic Interaction (HumanIT): A study protocol for a clinical trial evaluating brain neuroplasticity and functional performance after lower limb loss ’’, which we would like to have considered for publication in ‘PLOS ONE” as a protocol. 

We would like to thank the reviewers for providing valuable recommendations that were helpful for improving the quality of our manuscript. The suggestions of the reviewers have been incorporated in the revised version of our manuscript as much and as closely as possible, while maintaining the journal’s guidelines. 

Enclosed you can find a detailed description of our adaptations in the form of a point-by-point response to the reviewer’s recommendations. All changes to the original manuscript are indicated by track changes. 

Thank you for considering this paper for publication.

Sincerely yours,

On behalf of all co-authors,

Elke Lathouwers

 

Reviewer #1: 

This is a study protocol for a clinical trial that will evaluate brain neuroplasticity and functional performance after lower limb loss. The research questions are clearly stated and look sound to me, the sample size computations rely on reasonable assumptions, limitations of the study are listed. My overall impression is therefore positive. However, I list below some points that should be clarified.

1. Allocation to treatment arms. It is not clear whether allocation to treatments will be random or not. Please clarify.

Answer: We agree with the reviewer that the allocation was not clear. The allocation will be pseudo-randomised since we will perform alternating allocation. To clarify we adapted lines 139-140 as follows: 

 “A total of 40 participants with a lower limb amputation is required and will be allocated alternatingly (i.e., pseudo-randomised) to one of the two prosthetic groups.”

2. Data permissions. It is not clear whether participants will be asked to agree on making deidentified data available to the public (see also point no. 3).

Answer: We greatly appreciate your attention to the issue of data permissions and participants' agreement to make deidentified data publicly available. We want to assure you that the informed consent document for our study addresses this specific concern. Section 9.3 of the informed consent form, covers this concern and states that information might be shared with other researchers. More information on the requisites to obtain the deidentified data is provided in point no. 3. 

“Your participation in the trial means that your personal data

- are collected by the investigator, and

- are used in an encoded form by the trial sponsor. 

The investigator and the sponsor can only use the encoded personal data for research purposes in connection with scientific publications within the context of the trial that you participate in, or for a broader use of the encoded data if described below.

In addition, the sponsor may provide access to the encoded data to external researchers (that are not involved in this trial). In the event an external researcher wants to use the data in a project not yet described in this document, this project will have to be approved by an Ethics Committee. If your encoded trial data are sold, you will not benefit from this.”

To clarify the consent on making deidentified data available in the manuscript we added lines 263-265: 

“By signing the informed consent, participants also provide their agreement that encoded data may be shared with other researchers beyond the scope of the current study”. 

3. Data availability. It is good, and consistent with the policy of the journal, that the authors wish to make deidentified data publicly available. That the data managers would reserve the right to refuse data access is instead a bit too generic. The authors should clarify the criteria that will be used to refuse access to the data. The public repository where the data will be stored should also be declared.

Answer: The data will be made available upon reasonable request. Interested parties seeking access to the data are required to contact the corresponding author via email, with a copy sent to both the Data Protection Officer of the Vrije Universiteit Brussel and the Medical Ethics Commission of Vrije Universiteit Brussel and UZ Brussel. This correspondence must encompass the researcher's background, study protocol, intended utilization of the data, data handling protocol, and details of ethical clearance. Subsequently, the Medical Ethics Commission of the Vrije Universiteit Brussel and UZ Brussel will undertake a thorough review of the application, offering their professional advice. A decision will be rendered, with due notification to the applicant. In the event of data access denial, a substantiated rationale will be communicated.

We added these procedures to the manuscript lines 287-296:

“Interested parties seeking access to the data are required to contact the corresponding author (Kevin.De.Pauw@vub.be) via email, with a copy sent to both the Data Protection Officer (DPO@vub.be) of the Vrije Universiteit Brussel and the Medical Ethics Commission of Vrije Universiteit Brussel and UZ Brussel (ethiek@uzbrussel.be). This correspondence must encompass the researcher's background, study protocol, intended utilization of the data, data handling protocol, and details of ethical clearance. Subsequently, the Medical Ethics Commission of the Vrije Universiteit Brussel and UZ Brussel will undertake a thorough review of the application, offering their professional advice. A decision will be rendered, with due notification to the applicant. Should access be granted, the applicant will also receive access to the server location where a copy of the encoded data is stored. In the event of data access denial, a substantiated rationale will be communicated.”

 

4) Statistical analysis. The sentence “methods … will include multivariate and univariate analyses and t-tests” is a bit too generic. First, the longitudinal structure of the data should be accounted for (data will be in the form of repeated measurements). Second, differences will be affected by confounding factors (age, gender, BMI, …) and will need to be adjusted (see also point no 5). As a result, mixed-effects regression models should be considered to analyze the data.

Answer: We agree that the statistical plan is a bit too generic. We have a repeated measure design with multiple outcome variables. In this case, a mixed linear model would indeed be preferred. If the assumption to perform a linear mixed model would not be fulfilled, we will fit a generalized linear mixed model. If those assumptions would also not be fulfilled, we will consider multivariate and univariate analyses or non-parametric equivalents. With regard to the confounding variables, we will include confounding variables (age, handedness, sex, side of amputation, amount of daily prosthetic use and the average amount of daily physical activity) to our regression model. The amount of confounding factors that can be eventually included in our analyses will highly depend on the model building itself and the sample size we will reach. To add this nuance to the manuscript we altered the statistical paragraph lines 235-244 as follows: 

“Statistical analyses will be performed using R(1), MRtrix(2), and the hMRI toolbox(3). Statistical methods for brain neuroplasticity and functional performance testing outcomes will include mixed linear or generalised linear modelling depending on the model assumptions being fulfilled. If those analyses cannot be conducted, assumptions will be checked to perform multivariate or univariate analyses. If the assumptions for those types of analyses are also not fulfilled, non-parametric equivalents will be applied. The data will be adjusted for confounding variables like age, sex, handedness, amount of daily prosthetic use and the average amount of daily physical activity. The biomechanical gait parameters will be compared within and between groups using statistical parametric or non-parametric mapping. The statistical significance level will be set at 5% for all analyses.”

5) While the list of primary and secondary outcomes is well documented, the protocol lacks information about auxiliary subject information (age, gender, BMI, …) that will be collected.

Answer: We thank the reviewer for noting that the auxiliary subject information was lacking. We added a paragraph to the method section lines 181-190 describing all auxiliary data that will be collected:

“Auxiliary subject information will be collected during the first day of data collection for the participants with an amputation. The subject information that will be collected comprises age, sex, height, length, weight, handedness, average amount of daily physical activity, date of amputation, side of amputation, stump length, comorbidities, use of medication, presence of phantom pains, history of trauma and the history of falls. During the follow-up measurements, we will again perform the anamnesis and extend this with questions on the rehabilitation process, if other prosthetic devices were tested, and the duration of prosthesis use throughout the day. Auxiliary subject information that will be gathered during the first day of data collection within the control group of able-bodied participants will comprise age, sex, height, length, weight, handedness and average daily physical activity.”

Reviewer #2: 

This study protocol outlines the experimental design for a clinical trial aimed at evaluating the neuroplastic, functional, quality of life, and biomechanical effects associated with the use of lower limb transtibial prostheses featuring a passive articulated ankle. The results will be compared to control cohorts, one using a standard SACH-style ankle and another comprising an able-bodied group. Overall, the manuscript is well-written, and the experimental design is sound. However, I have a few concerns that should be addressed prior to consideration for publication.

1. While the authors provide evidence that the neuroplastic changes related to prosthetic ankle choice remain poorly understood, it was not clear to me how novel the functional, quality of life, and biomechanical assessments are. I suggest that the authors provide further discussions around the existing literature and describe how the data collected in their two secondary aims will contribute to the knowledge in the field. I recommend considering a more focused approach, such as evaluating correlations across numerous outcome parameters. Might I suggest that the author consider restricting their aims to a single a, perhaps evaluating correlations across their numerous outcome parameters. For example, the objective of the protocol may be stronger if it were about examining neuroplasticity and how these changes may relate to functional, QoL, and biomechanical outcomes. It certainly would be interesting to better understand how biomechanics and neuroplasticity may be related, etc.

Answer: We thank the reviewer for these suggestions. We discussed this topic within the authors‘ team. Based on our systematic review on therapeutic benefits of ankle-foot prostheses, we identified the lack at mid- and longer term studies comparing prosthetic ankle devices using biopsychological measurements and the lack at studies comparing prosthetic devices on neuroplasticity(4). We recognize the importance of addressing this gap, and we want to emphasize that our study aims to directly bridge the link between this identified gap in the literature and the current state of research. For this reason, and after careful consideration, we've determined that our primary focus should remain on addressing our original objectives.

2. The MRI analysis is central to the first objective of this study. However, when describing this outcome measure (line 186), there is very little information about how the collected MRI data will be analyzed. I suggest that the authors further develop and describe a plan for the handling and analysis of MRI data. For instance, they could elaborate on the specific measures, brain regions, networks, temporal changes, etc., that will be interrogated, and clarify how they plan to quantify the degree of plasticity demonstrated.

Answer: In response to your suggestion, we have provided a more detailed description of the MRI data analysis lines 192 – 214:

“Neuroplasticity can be defined as a reorganisation of brain structure, function and connectivity within the central nervous system in response to intrinsic and extrinsic stimuli(5). Based on our hypotheses, we are mainly interested in, but not limited to, the motor-related regions of the brain, given the exploratory nature of this study (6). The outcome parameters include connectivity and white and grey matter microstructure measures (e.g., characteristic path length, clustering coefficient, efficiency, neurite density index, fiber density, and fiber cross-section, fiber density cross-section).

Pre- and post-intervention MRI scans will be acquired supine on a 3-Tesla MRI unit using a 48-channel head coil. A 30-minute protocol will be followed, comprising scout images, 3D T1-weighted spin-echo images, diffusion-weighted imaging, resting state-fMRI, and 3D-QALAS. An anatomical 3D T1-weighted image and a multi-shell diffusion-weighted dataset will be acquired, as well as a resting-state fMRI acquisition and a 3D-QALAS acquisition. The diffusion dataset includes several b0 images, and for both the diffusion and resting state data, additional data using reversed-phase directions will be acquired. The 3D-QALAS acquisition allows for a simultaneous T1, T2 and proton density mapping, additional data using reversed-phase directions will be acquired, making it sensitive to microstructural changes in the brain (7, 8). 

After data acquisition, the MRI data will be pre-processed, and the outcome parameters will be computed. FSL will be used to pre-process the diffusion-weighted imaging and resting state-fMRI data(9, 10, 11). Pre-processing consists of denoising, distortion, eddy current and bias field correction, and motion correction with outlier replacement. Afterward, a MRtrix3 performs a fixed-based analysis in the white matter regions, while a NODDI analysis is done in the grey matter areas(12, 13). To analyse the 3D-QALAS data, SyMRI will be used (SyntheticMR AB, Linköping, Sweden), after which grey matter, white matter and myelin maps are estimated from the relaxometry maps(14, 15). Once the relaxometry and tissue maps are obtained, a voxel-based analysis will be performed using a modified version of the hMRI toolbox of SPM12(16, 17), using MATLAB.”

The quantification of neuroplasticity will be conducted by assessing differences in the outcome parameters through the application of mixed linear regression. A description of this approach is provided in the statistical analysis section lines 236-245:

“Statistical analyses will be performed using R(1), MRtrix(2), and the hMRI toolbox(3). Statistical methods for brain neuroplasticity and functional performance testing outcomes will include mixed linear or generalised linear modelling depending on the model assumptions being fulfilled. If those analyses cannot be conducted, assumptions will be checked to perform multivariate or univariate analyses. If the assumptions for those types of analyses are also not fulfilled, non-parametric equivalents will be applied. The data will be adjusted for confounding variables like age, sex, handedness, amount of daily prosthetic use and the average amount of daily physical activity. The biomechanical gait parameters will be compared within and between groups using statistical parametric or non-parametric mapping. The statistical significance level will be set at 5% for all analyses.“

3. Lines 103-106 briefly describe a prior systematic review. It would strengthen the manuscript to summarize what the literature highlighted in this section found and how it relates to this study protocol.

Answer: In response to your feedback, we have included a concise summary of the systematic review. We believe this addition enhances the manuscript by providing a clear link between the existing literature and the objectives of our study. Lines 90-101 now read: 

“Understanding the influence of passive and articulated passive prostheses on brain neuroplasticity will provide vital information on the underlying mechanisms in the context of fitting an individual with amputation with the most beneficial prosthetic device to improve functionality, reduce fall risk and ultimately improve quality of life. A recent systematic review compared the therapeutic benefits of different ankle prostheses (i.e., passive, quasi-passive and active) during daily activities(4). This review demonstrated that although many short-term benefits of advanced prosthetic devices have been shown through user evaluations, there is a lack in the literature concerning mid-to-long-term holistic assessments employing a psychophysiological approach(4). Furthermore, among the included manuscripts, only De Pauw et al. investigated the difference in brain dynamics between two types of prosthetic ankles by means of EEG(18). Nevertheless, research on the impact of the prosthesis type on neuroplasticity is notably absent from scientific literature. Therefore, our study aims to research brain neuroplasticity induced by different types of prosthetic ankle devices.” 

References

1. R Development Core Team. R: A Language and Environment for Statistical Computing. R Foundation for Statistical Computing. 2019 [

2. Tournier JD, Smith R, Raffelt D, Tabbara R, Dhollander T, Pietsch M, et al. MRtrix3: A fast, flexible and open software framework for medical image processing and visualisation. NeuroImage. 2019;202:116137.

3. Tabelow K, Balteau E, Ashburner J, Callaghan MF, Draganski B, Helms G, et al. hMRI – A toolbox for quantitative MRI in neuroscience and clinical research. NeuroImage. 2019;194:191-210.

4. Lathouwers E, Díaz MA, Maricot A, Tassignon B, Cherelle C, Cherelle P, et al. Therapeutic benefits of lower limb prostheses: a systematic review. Journal of NeuroEngineering and Rehabilitation. 2023;20(1):4.

5. Puderbaugh M, Emmady PD. Neuroplasticity. StatPearls. Treasure Island (FL): StatPearls Publishing Copyright © 2023, StatPearls Publishing LLC.; 2023.

6. Molina-Rueda F, Navarro-Fernández C, Cuesta-Gómez A, Alguacil-Diego IM, Molero-Sánchez A, Carratalá-Tejada M. Neuroplasticity Modifications Following a Lower-Limb Amputation: A Systematic Review. Pm r. 2019;11(12):1326-34.

7. Fujita S, Hagiwara A, Hori M, Warntjes M, Kamagata K, Fukunaga I, et al. Three-dimensional high-resolution simultaneous quantitative mapping of the whole brain with 3D-QALAS: An accuracy and repeatability study. Magn Reson Imaging. 2019;63:235-43.

8. Kvernby S, Warntjes MJ, Haraldsson H, Carlhäll CJ, Engvall J, Ebbers T. Simultaneous three-dimensional myocardial T1 and T2 mapping in one breath hold with 3D-QALAS. J Cardiovasc Magn Reson. 2014;16(1):102.

9. Jenkinson M, Beckmann CF, Behrens TE, Woolrich MW, Smith SM. FSL. Neuroimage. 2012;62(2):782-90.

10. Smith SM, Jenkinson M, Woolrich MW, Beckmann CF, Behrens TE, Johansen-Berg H, et al. Advances in functional and structural MR image analysis and implementation as FSL. Neuroimage. 2004;23 Suppl 1:S208-19.

11. Woolrich MW, Jbabdi S, Patenaude B, Chappell M, Makni S, Behrens T, et al. Bayesian analysis of neuroimaging data in FSL. Neuroimage. 2009;45(1 Suppl):S173-86.

12. MRtrix. Fibre density and cross-section - Single-tissue CSD - MRtrix 3.0 documentation [Available from: https://mrtrix.readthedocs.io/en/dev/fixel_based_analysis/st_fibre_density_cross-section.html.

13. Zhang H, Schneider T, Wheeler-Kingshott CA, Alexander DC. NODDI: practical in vivo neurite orientation dispersion and density imaging of the human brain. Neuroimage. 2012;61(4):1000-16.

14. Hagiwara A, Warntjes M, Hori M, Andica C, Nakazawa M, Kumamaru KK, et al. SyMRI of the Brain: Rapid Quantification of Relaxation Rates and Proton Density, With Synthetic MRI, Automatic Brain Segmentation, and Myelin Measurement. Invest Radiol. 2017;52(10):647-57.

15. West J, Warntjes JB, Lundberg P. Novel whole brain segmentation and volume estimation using quantitative MRI. Eur Radiol. 2012;22(5):998-1007.

16. Friston KJ, Glaser DE, Henson RN, Kiebel S, Phillips C, Ashburner J. Classical and Bayesian inference in neuroimaging: applications. Neuroimage. 2002;16(2):484-512.

17. Tabelow K, Balteau E, Ashburner J, Callaghan MF, Draganski B, Helms G, et al. hMRI - A toolbox for quantitative MRI in neuroscience and clinical research. Neuroimage. 2019;194:191-210.

18. De Pauw K, Cherelle P, Tassignon B, Van Cutsem J, Roelands B, Marulanda FG, et al. Cognitive performance and brain dynamics during walking with a novel bionic foot: A pilot study. PLOS ONE. 2019;14(4):e0214711.

---

## [Editor Report · Decision Letter 1]

19 Feb 2024

Human-Prosthetic Interaction (HumanIT): A study protocol for a clinical trial evaluating brain neuroplasticity and functional performance after lower limb loss

PONE-D-23-27271R1

Dear Dr. De Pauw,

We’re pleased to inform you that your manuscript has been judged scientifically suitable for publication and will be formally accepted for publication once it meets all outstanding requirements.

The reviewer requests were mostly appropriately answered. However, one pending issue remains **data availability**, which appears to hinge upon availability of the corresponding author. It is also unclear where and how the data will be stored to ensure lasting possibility for access. Regarding this concern, the PLOS ONE team will be following up to ensure the data availability is compliant with PLOS ONE's data policies.

For the plagiarism concern, a publications assistant reviewed the automatically generated plagiarism report and could not find issues, so this has been resolved.

Kind regards,

Heike Vallery

Academic Editor

PLOS ONE

Additional Editor Comments (optional):

See above concerning data availability

Reviewers' comments:

The manuscript was not reviewed again by external reviewers.

---

## [Editor Report · Acceptance letter]

11 Mar 2024

PONE-D-23-27271R1 

PLOS ONE

Dear Dr. De Pauw, 

I'm pleased to inform you that your manuscript has been deemed suitable for publication in PLOS ONE. Congratulations! Your manuscript is now being handed over to our production team.

Kind regards, 

on behalf of

Dr. Heike Vallery 

Academic Editor

PLOS ONE